# Entropy of Simulated Liquids Using Multiscale Cell Correlation

**DOI:** 10.3390/e21080750

**Published:** 2019-07-31

**Authors:** Hafiz Saqib Ali, Jonathan Higham, Richard H. Henchman

**Affiliations:** 1Manchester Institute of Biotechnology, The University of Manchester, 131 Princess Street, Manchester M1 7DN, UK; 2School of Chemistry, The University of Manchester, Oxford Road, Manchester M13 9PL, UK

**Keywords:** structure, thermodynamics, probability distribution, force, torque, coordination, conformation, molecular dynamics simulation

## Abstract

Accurately calculating the entropy of liquids is an important goal, given that many processes take place in the liquid phase. Of almost equal importance is understanding the values obtained. However, there are few methods that can calculate the entropy of such systems, and fewer still to make sense of the values obtained. We present our multiscale cell correlation (MCC) method to calculate the entropy of liquids from molecular dynamics simulations. The method uses forces and torques at the molecule and united-atom levels and probability distributions of molecular coordinations and conformations. The main differences with previous work are the consistent treatment of the mean-field cell approximation to the approriate degrees of freedom, the separation of the force and torque covariance matrices, and the inclusion of conformation correlation for molecules with multiple dihedrals. MCC is applied to a broader set of 56 important industrial liquids modeled using the Generalized AMBER Force Field (GAFF) and Optimized Potentials for Liquid Simulations (OPLS) force fields with 1.14*CM1A charges. Unsigned errors versus experimental entropies are 8.7 J K${}^{-1}$ mol${}^{-1}$ for GAFF and 9.8 J K${}^{-1}$ mol${}^{-1}$ for OPLS. This is significantly better than the 2-Phase Thermodynamics method for the subset of molecules in common, which is the only other method that has been applied to such systems. MCC makes clear why the entropy has the value it does by providing a decomposition in terms of translational and rotational vibrational entropy and topographical entropy at the molecular and united-atom levels.

## 1. Introduction

Molecular liquids are present in numerous systems in chemistry and biology. However, methods to calculate their entropy are scarce or limited in scope. Entropy quantifies the probability distribution of quantum states of a system and, together with energy, determines a system’s stability. The most common route used to determine entropy is indirect, being as a difference with respect to a reference state, typically the ideal gas or a non-interacting set of atoms. The entropy difference may be extracted from integrated heat capacity changes or from the Gibbs energy difference, either as its temperature derivative or as a difference with enthalpy [1]. While there is a range of methods to compute entropy [2,3,4,5,6,7,8,9,10], those that use single molecular dynamics or Monte Carlo simulations are advantageous because of the ease of using standard simulation methods and because such approaches directly yield and explain entropy and structure in terms of the full probability distribution of the system of interest. However, because the ensemble of molecular configurations generated by standard simulation methods is only a tiny fraction of the full ensemble corresponding to a system’s entropy, special techniques are required to extrapolate to the full probability distribution and entropy.

The probability distributions to evaluate entropy are typically over the coordinates of the system, which may be Cartesian coordinates, bonds-angles-dihedrals, or interatomic distances. Histogram-based methods, because of their arbitrary bin-widths, can only give the entropy difference relative to a reference, which is typically the uniform distribution. Even then, the entropy difference may be unrealistic for strongly interacting systems such as those with covalent bonds because of the omission of quantum effects which necessarily keep the entropy non-negative. For this reason, histogram methods are often restricted to softer degrees of freedom such as dihedrals or atomic distances. The simplest approach ignores coordinate correlations by considering each coordinate separately, for example, in dihedral angles [11]. Higher-order correlations can be included such as the radial distribution function [12,13,14] or a mutual-information expansion [15,16] but at greater computational expense and complexity, even for second-order, although some correlations are small and can be excluded [17,18,19]. Extensions to higher orders are difficult and do not necessarily lead to more accuracy [15,20]. Mutual information in terms of discrete rotamers has been found to converge much faster, enabling up to eighth order [21]. An alternative strategy for high-dimensional data sets is the k-Nearest Neighbours method [16,22,23,24] which more adaptively estimates density from the distances between configurations but at the price of having many distances to compute and still requiring a lot of data to converge.

Significant simplification of the theory, greater speed of convergence and a route to the direct calculation of entropy is provided by assuming a multivariate Gaussian probability distribution [25]. Entropy is directly computed from the quantum states of the set of harmonic-oscillator eigenvectors [26,27]. The main limitation of the method is the suitability of the Gaussian distribution, given that typical potential energy surfaces for flexible molecules [28] or liquids [27,29] have multiple minima, compounded by the difficulty of how to specify the minima. A hybrid solution to this problem is to replace the diagonal elements of the coordinate covariance matrix with the entropy of the probability distributions [30,31]. Another solution is to incorporate multiple Gaussians [32]. An approach particularly relevant to the case of liquids is the 2-Phase Thermodynamics (2PT) method, which calculates entropy from the spectrum of vibrational frequencies derived from the velocity auto-correlation function and the gas-phase fluidicity [33]. Another viable method for liquid-phase entropy is the cell approximation which maps regions of the potential energy surface into single, representative energy wells, whose entropy is determined from the force [34] plus an entropy term for the probability distribution of the energy wells [35]. This is the method we have been working to generalise, progressing from liquid argon [34] to liquid water with its rotational vibration and orientational degrees of freedom [35,36,37], organic liquids with an internal one-dimensional dihedral entropy [38], single molecules with internal entropy based on force correlation [39], and molecular liquids in a multiscale framework from atom to united atom to molecule to system [40]. This development has been supported by extensive parallel studies on the entropy of aqueous solutions [41,42,43,44,45,46,47]. With the main ideas now in place to make the method general, to encapsulate the main features of the method we name it Multiscale Cell Correlation (MCC).

Here we extend MCC to calculate the entropy of 56 important industrial liquids. These represent a class of system which no other method has been capable of calculating entropy except for the 2PT method, which has been tested on a smaller subset of 14 liquids [48], argon [33], water [49], carbon dioxide [50], and methanol and hexane including torsional fluidicity [51]. The first improvement here in MCC is a more appropriate application of the mean-field cell approximation to the weakly correlated non-bonded and dihedral degrees of freedom and not to the correlated bonded and angular degrees of freedom as had been done in previous work [39,40]. Strong correlations for the bonded atoms invalidate the cell approximation and can be accounted for in the force covariance matrices. Related to this, force and torque covariances are evaluated separately because of their weak correlation [40]. The second key improvement is a new way to account for correlation between dihedrals by using a covariance matrix of conformation correlation, a method that scales with the square of the number of dihedrals. The 56 liquids are tested using two force fields: OPLS (Optimized Potentials for Liquid Simulations) with 1.14*CM1A charges [52] and GAFF [53] (Generalized AMBER Force Field), for both of which parameters can be generated in an automated fashion for a wide range of molecules. A decomposition of the entropy in six terms gives an insightful and intuitive explanation of why molecules have the entropy they do. Compared to our earlier study in which a comparison with 2PT was inconclusive because there were few liquids in common, MCC is found to be significantly closer to experiment than 2PT, which in most cases underestimates experiment. An analysis of entropy components suggests that the internal entropy of 2PT is responsible for this underestimation, even when torsional fluidicity is included [51]. The findings show that MCC is well placed to scale to complex multi-component systems with multiple length scales.

## 2. Theory

### 2.1. Entropy Decomposition

The entropy of molecular liquids is well captured at two different length scales [40]: the molecule (M) level and the united-atom (UA) level. A united atom is defined here as a non-hydrogen atom together with any bonded hydrogen atoms and is taken as a rigid body with both translational and rotational degrees of freedom rather than only translation as for a point-particle unless there are no hydrogens. Such an approach captures softer collective dihedral motion of hydrogens while ignoring their individual stretching and bending motions which have negligible entropy, owing to the low mass of hydrogen and its higher bond and angle vibrational frequencies. At the other extreme of the whole system, the entropy of its three translational and three rotational degrees of freedom is negligible on a per-molecule basis. Coordinate systems at the molecule and united-atom levels are defined as before [40]. For a molecule this is its three principal axes with the origin at the centre of mass. For a united atom the axes and centre of mass depend on the number of bonded united and hydrogen atoms. All non-linear molecules and united atoms have three translational degrees of freedom. Linear molecules in terms of their united atoms or linear united atoms in terms of their hydrogens have two rotational degrees of freedom. United atoms with no hydrogens have no rotational degrees of freedom.

In the cell approximation the potential energy surface is partitioned into energy wells, and in the multiscale approximation this partitioning is done at the molecule and united-atom levels. This brings about two kinds of entropy term: vibrational relating to the average size of the energy wells, termed a cell, and topographical relating to the probability of the energy wells. The vibrational term at each level is further partitioned according to the translational (transvib) and rotational (rovib) degrees of freedom. The translational component of the topographical entropy at the molecular level is zero for a pure liquid because exchanging identical molecules leads to no change. The rotational topographical entropy (topo) at the molecule level is termed the orientational entropy. At the united-atom level the translational topographical entropy is the conformational entropy, while the rotational component, corresponding to hydrogen-bond arrangements, is negligible for the liquids studied here. The total entropy per molecule for a liquid is therefore taken as the sum of six terms
(1)$$\begin{array}{c} S_{\mathrm{total}}=S_{M}^{\mathrm{transvib}}+S_{M}^{\mathrm{rovib}}+S_{M}^{\mathrm{topo}}+S_{\mathrm{UA}}^{\mathrm{transvib}}+S_{\mathrm{UA}}^{\mathrm{rovib}}+S_{\mathrm{UA}}^{\mathrm{topo}} \end{array}$$

### 2.2. Molecular Vibrational Entropy

All four vibrational entropy terms are calculated in the harmonic approximation using the equation for a collection of $N_{\mathrm{vib}}$ quantum harmonic oscillators
(2)$$S^{\mathrm{vib}}=k_{B}\;\sum_{i=1}^{N_{\mathrm{vib}}}\left(\frac{h\nu_{i}/k_{B}T}{e^{h\nu_{i}/k_{B}T}-1}-\ln\left(1-e^{-h\nu_{i}/k_{B}T}\right)\right)$$
where $k_{B}$ is Boltzmann’s constant, *h* is Planck’s constant, *T* is temperature and $\nu_{i}$ are the vibrational frequencies. Different to previous work [40], translational and rotational vibrational entropy are evaluated separately, justified by the absence of correlations between the forces and torques that are used to evaluate them. For $S_{M}^{\mathrm{transvib}}$, $N_{\mathrm{vib}}=3$ and $\nu_{i}$ are calculated using [39,40]
(3)$$\nu_{i}=\frac{1}{2\pi }\sqrt{\frac{\lambda_{i}}{k_{B}T}}$$
where $\lambda_{i}$ are the eigenvalues of the 3×3 mass-weighted force covariance matrix of the molecule with elements $\langle F_{i}^{\prime }F_{j}^{\prime }\rangle$, with *i* and *j* ranging over the three axes x,y,z and averaging over all molecules in all simulation frames. Mean-field, mass-weighted forces are defined as $F_{i}^{\prime }=F_{i}/\left(2\sqrt{m}\right)$ where *m* is the molecule’s mass, and $F_{i}$ is half the component of net force on all the atoms of the molecule rotated into the molecule’s coordinate frame. In practice, this matrix is essentially diagonal because forces along different axes are negligibly correlated. The halving is done in the mean-field cell approximation [34,35] whereby every pairwise energy term and therefore its negative coordinate derivative, the force, is partitioned equally between the atoms involved. The mean-field cell approximation is justified in liquids because average molecular energies and forces in many-body systems are weakly correlated with the position of any other neighbouring molecule. Only over the short duration of a repulsive collision is the correlation significant. To calculate $S_{M}^{\mathrm{rovib}}$ with Equation (Equation 2), $N_{\mathrm{vib}}=3$ unless the molecular is linear with respect to its united atoms, in which case $N_{\mathrm{vib}}=2$. The vibrational frequencies $\nu_{i}$ are calculated using Equation (Equation 3) with eigenvalues from the $N_{\mathrm{vib}}\times N_{\mathrm{vib}}$ moment-of-inertia-weighted torque covariance matrix of the molecule, whose elements are $\langle \tau_{i}^{\prime }\tau_{j}^{\prime }\rangle$, where $\tau_{i}^{\prime }=\tau_{i}/\left(2\sqrt{I_{i}}\right)$ for each axis i=x,y,z and $I_{i}$ is the respective moment of inertia, with torque halving being done as for the forces.

### 2.3. United-Atom Vibrational Entropy

The procedure at the united-atom level to evaluate $S_{\mathrm{UA}}^{\mathrm{transvib}}$ and $S_{\mathrm{UA}}^{\mathrm{rovib}}$ in Equation (Equation 1) is similar to that at the molecule-level but with some differences. United atoms are used in place of molecules to evaluate the forces, torques, masses and moments of inertia. $N_{\mathrm{vib}}$ in Equation (Equation 2) for united-atom translation equals 3N−6, where *N* is the number of united atoms and the six vibrations removed correspond to the six largest eigenvalues which are already accounted for as molecular translation and rotation. $N_{\mathrm{vib}}$ in Equation (Equation 2) for united-atom rotation depends on the number of non-linear, linear and point united atoms, as well as the linearity of the whole molecule. Non-linear and linear united atoms contribute 3 and 2 degrees of freedom, respectively, and the largest six or five eigenvalues are removed if the molecule is non-linear or linear. A notable difference compared to the molecule level is that the mean-field cell approximation is not made for bonded atoms or bonded 1–3 interactions corresponding to angles. The forces of such atoms are strongly correlated, a correlation that is accounted for in the covariance matrix. However, the mean-field approximation is still made for united-atom rotation and dihedral vibration whose correlations with neighbours are weak relative to the overall torque or force or which largely average out to zero because of averaging in different reference frames. Consequently, forces in the united-atom matrix are not halved but united-atom torques are halved. To implement the cell approximation for dihedrals, the $N_{\mathrm{dih}}$ lowest eigenvalues of the united-atom force covariance matrix are halved twice (force-squared), where $N_{\mathrm{dih}}$ is the number of united-atom dihedrals, because these eigenvalues correspond to the soft conformational eigenvectors.

### 2.4. Molecular Topographical Entropy

The molecular topographical entropy $S_{M}^{\mathrm{topo}}$ in Equation (Equation 1) only has a rotational contribution for a pure liquid, referred to as the orientational entropy. Based on the idea that neighbouring molecules discretize a molecule’s rotational motion, $S_{M}^{\mathrm{topo}}$ is estimated using an average of the number of orientations weighted by the probability $p(N_{c})$ of molecular coordination number $N_{c}$ using [40]
(4)$$\begin{array}{c} S_{M}^{\mathrm{topo}}=k_{B}\sum_{N_{c}}p\left(N_{c}\right)\ln\left[\max\left(1,{\left(N_{c}^{3}\pi \right)}^{1/2}/\sigma \right)\right] \end{array}$$
where σ is the symmetry number of the molecule according to its united atoms. The max function only takes effect for the very small values of $N_{c}$ which are rare. Thus there are $∼N_{c}^{1/2}$ orientations per rotational axis, and every orientation is taken to have the same probability, $\sigma /{\left(N_{c}^{3}\pi \right)}^{1/2}$, justified by the weak correlation of these moderately polar molecules with their neighbours. For linear molecules with two axes of rotation [40], the equation is
(5)$$\begin{array}{c} S_{M}^{\mathrm{topo}}=k_{B}\sum_{N_{c}}p\left(N_{c}\right)\ln\left[\max\left(1,N_{c}/\sigma \right)\right] \end{array}$$

Molecules with a single united atom may still have orientational entropy at the atom-level if their hydrogens sufficiently break symmetry, so as to form distinct energy wells. Ammonia is included in this category, as water had been earlier [40], but methane and hydrogen sulfide are not. $N_{c}$ is evaluated using the parameter-free relative angular distance (RAD) method [54,55] according to the centre of mass of each molecule. RAD determines $N_{c}$ from a single configuration in good agreement with those using a cut-off at the first minimum in the radial distribution function. It avoids the need for a mean-field, spherically-symmetric cut-off that must either be chosen arbitrarily or evaluated from the pre-computed radial distribution function.

### 2.5. United-Atom Topographical Entropy

The topographical entropy at the united-atom level, $S_{\mathrm{UA}}^{\mathrm{topo}}$, also called the conformational entropy, is derived from the distribution of discrete conformations for all flexible dihedrals involving united atoms. Unlike in the previous work on liquid entropy [40] in which the molecules only had a maximum of one flexible dihedral, a number of molecules here have multiple dihedrals. Given that they may be correlated, we present a new method to account for this using a conformation correlation matrix. Each molecule has $N_{\mathrm{dih}}$ dihedrals, taken as four consecutive, bonded united atoms. The topographial entropy of dihedrals at the atomic level and involving hydrogen are ignored, either because they have only one conformation by symmetry, such as a methyl group, or because they have negligibly more than one conformation, such as a hydroxyl, owing to limited variable hydrogen-bonding capability to neighbour molecules. The molecules considered here have three conformations per dihedral: trans (*t*), gauche− (g−) and gauche+ (g+), defined with boundaries in dihedral angle at 120${}^{\circ }$, 0${}^{\circ }$ and −120${}^{\circ }$, respectively. Thus each molecule has available $3N_{\mathrm{dih}}$ conformations. Every combination of conformations for each molecule is termed a conformer, and the total possible number of conformers is $3^{N_{\mathrm{dih}}}$. Overall we ensure there is no double-counting of identical conformers by treating g− and g+ as distinct and dividing by the rotational symmetry number in Equations (Equation 4) and (Equation 5). We construct the $3N_{\mathrm{dih}}\times 3N_{\mathrm{dih}}$ correlation matrix ρ which has elements
(6)$$\begin{array}{c} \rho_{ij}=p_{ij}r_{ij}/P_{m(i)} \end{array}$$
where $p_{ij}$ is the probability of simultaneously having the conformation pair *i* and *j*, normalised such that $\sum_{i=3m}^{3m+2}\sum_{j=3n}^{3n+2}p_{ij}=1$ for the square sub-block over all conformation *i* and *j* of the respective dihedrals *m* and *n*, and $r_{ij}$ is the Pearson correlation coefficient of conformations *i* and *j*, given by
(7)$$\begin{array}{c} r_{ij}=\frac{p_{ij}-p_{i}p_{j}}{{\left[\left(p_{i}-p_{i}^{2}\right)\left(p_{j}-p_{j}^{2}\right)\right]}^{1/2}} \end{array}$$
where $p_{i}$ is the probability of conformation *i*. Note that $r_{ii}=1$, $p_{ii}=p_{i}$, and $p_{ij}=0$ if *i* and *j* belong to the same dihedral. $P_{m(i)}$ in Equation (Equation 6) are normalisation constants, one per dihedral *m*, that are defined to ensure $\sum_{i=3m}^{3m+2}\sum_{j=1}^{3N_{\mathrm{dih}}}\rho_{ij}=1$ for each dihedral *m*. Thus $\rho_{ij}$ represents the fraction of correlation that conformation pair ij makes to the total correlation that *i*’s dihedral *m* has with all conformations of all dihedrals. Similar to the von Neumann entropy of the density matrix [56], the total conformational entropy is given by
(8)$$S_{\mathrm{UA}}^{\mathrm{topo}}=-k_{B}\sum_{i=1}^{3N_{\mathrm{dih}}}\lambda_{i}\ln\left(\lambda_{i}\right)$$
where $\lambda_{i}$, the eigenvalues of ρ, are the probability of each conformer eigenvector, and each conformer eigenvector itself comprises the probabilities of each conformation. Unlike the density matrix, whose trace equals 1 [56], the trace of ρ ranges from 1, corresponding to full correlation between conformations, to a maximum of $N_{\mathrm{dih}}$, corresponding to fully uncorrelated conformations. For a molecule with uncorrelated conformations or with only one dihedral [40], its eigenvalues would be the diagonal elements of ρ and the conformer eigenvectors would be the individual conformations. At the other extreme of full correlation, as would occur when there is only one single conformer, one eigenvalue would equal 1 with its eigenvector being that very conformer, while the remaining eigenvalues would be zero. For cases of intermediate correlation between conformations, the eigenvector conformers would have various contributions from the correlated conformations, with entropy ranging from zero to the fully disordered value for all $N_{\mathrm{dih}}$ dihedrals.

### 2.6. Molecular Dynamics Simulations

The entropy was calculated for a series of 56 liquids using molecular dynamics simulations. All simulations were carried out with the sander module of the AMBER 14 simulation package [57]. Each system consists of 500 identical molecules in the liquid phase in a cubic box. The force fields used were GAFF [53] with AM1-BCC charges for all molecules and OPLS-AA with the 1.14*CM1A charges [52] for all molecules except acetonitrile, carbon dioxide, hydrogen sulfide and tetrafluoroethylene for which charges were not available on the LigParGen webserver [52]. In place of this for carbon dioxide, a simulation was run with the TraPPE (Transferable Potentials for Phase Equilibria) force field [58]. All molecules were built in standard geometry using xleap of AMBER 14. GAFF force-field parameters were generated with antechamber [59] and all molecules were placed in a cubic box of side 6 nm using Packmol [60]. For OPLS, the GROMACS topology and coordinate files were obtained by uploading a pdb of each molecule to the LigParGen webserver [52] with the 1.14*CM1A charges, the coordinates of the box of molecules were generated in GROMACS 5.1 [61], and the topology and coordinate files were converted into AMBER format using the AMBER ParmEd tool. Note that these OPLS charges differ to those in previous work [40] with the OPLS force field which had charges fitted to liquid-phase properties [62]. TraPPE parameters for carbon dioxide were added directly in by hand.

For equilibration, each system was minimized with 500 steps of steepest descent minimization, thermalized in a 100 ps molecular dynamics simulation at constant volume and temperature using a Langevin thermostat with a collision frequency of 5 ps${}^{-1}$, and brought to the correct density with 1 ns of molecular dynamics simulation at constant pressure using the Berendsen barostat with a time constant of 2 ps. For data collection, forces and coordinates were saved every 1 ps in a further 1 ns simulation under the same conditions, which earlier work had shown to be easily sufficient for converged values [40], in which as few as ten frames was often sufficient to achieve converged integer values in units of J K${}^{-1}$ mol${}^{-1}$. The pressure was 1 bar and the temperature was 298 K unless the liquid was gaseous at that temperature, in which case the boiling temperature at 1 bar was used as listed in Table 1. The exception is carbon dioxide, which does not liquefy at ambient pressure and so the pressure was set to 5.99 bar and temperature 220 K which is in the liquid-phase region, close to the triple point and matches conditions used in a 2PT study [50]. Simulations used SHAKE on all bonds involving hydrogen atoms, a non-bonded cutoff of 8 Å, periodic boundary conditions, particle-mesh Ewald summation with default parameters in AMBER, and a 2 fs timestep. Table 3 contains all the liquids simulated, for five of which the following abbreviations are used: dimethylformamide (DMFA), dimethylsulfoxide (DMSO), N-methyl acetamide (NMA), tert-butyl alcohol (TBA) and tetrafluoroethylene (TFE). Symmetry numbers in Equations (Equation 4) and (Equation 5) are listed in Appendix A. Entropies were calculated with in-house C++ and Perl code, reading in the force, coordinate and topology files and writing out eigenvalues and coordination numbers.

## 3. Results

### 3.1. Entropy Values

Figure 1 presents the entropy of 50 of the liquids calculated by MCC for the OPLS and GAFF force fields plotted against the respective experimental values [63,64,65,66,67,68,69,70]. Table 2 gives the mean unsigned and signed deviations, slopes, intercepts, Pearson correlation coefficients $R^{2}$, and zero-intercept slopes of entropies by the MCC and 2PT methods with respect to experiment. Table 3 contains the MCC entropy values for all 56 liquids, together with values from experiment, the MCC entropy of carbon dioxide with the TraPPE force field, and values using the 2PT method with the OPLS and GAFF force fields for fifteen liquids [48], carbon dioxide [50] and methanol and hexane including torsional fluidicity [51]. Statistical errors are negligible for the precision given.

The experimental entropies for most liquids were taken from the NIST Chemistry Webbook [63]. If more than one value was reported by different authors, all values were included, although for acetic acid, ethanol, ethylene glycol, formic acid, propanol and pyridine the spread is substantial, exceeding 10 J K${}^{-1}$ mol${}^{-1}$. Entropies were found elsewhere for ammonia [64], chloroform [65], methane [66], hydrogen peroxide [67], hydrogen sulfide [68] and carbon dioxide [69]. Values for ethylamine and triethylamine were calculated from the experimental gas-phase entropy, enthalpy of vaporization, and either heat capacity at constant pressure or partial pressure [63,70] (see Appendix A for details). For the remaining six liquids no values could be found in the literature. The experimental entropy is averaged if there is more than one value.

The entropy values calculated by MCC agree well with experiment, with Table 2 showing a mean unsigned error of less than 10 J K${}^{-1}$ mol${}^{-1}$, GAFF being slightly better than OPLS. The small mean signed errors, the slopes being marginally less than one, and the positive y-intercept suggest that MCC is slightly missing the dependence on molecular size, although forcing the line through zero brings about the correct unity slope. The excessive entropies seen for larger molecules in the earlier version of the theory [40] no longer occur because we no longer halve forces for hard internal degrees of freedom in the mean-field approximation.

Comparisons with experiment are affected by the accuracy of the force field. To compare MCC with 2PT, Table 2 contains the statistical quantities for the liquids studied by the 2PT method [48] listed in Table 3, comprising 14 with the GAFF force field [53] and 12 with the OPLS force field [62] together with the corresponding MCC values with GAFF and OPLS with 1.14*CM1A charges [52]. For both force fields, the mean unsigned error for 2PT is three times that of MCC, largely because the 2PT values are too small, shown by the negative mean error, negative y-intercept, and poorer correlation. The slope is close to unity but decreases when forced through the origin. The difference between the two methods is unlikely solely due to the different OPLS force fields, given the trend is present for GAFF, that the 2PT values using the earlier OPLS force field better reproduce liquid-phase entropy, and that the same trend was observed earlier when comparing with the same force field [40,71]. The variability in experiment for acetic acid and ethanol may affect this comparison, in that MCC is closer to the higher value and 2PT closer to the lower value, but this would be insufficient to affect the overall trend. The poorer MCC performance of OPLS with 1.14*CM1A charges compared to GAFF likely reflects the over-polarization of the charges to optimise their free energy of hydration [52]. This also likely explains the better performance of OPLS than GAFF for 2PT. Including the localized bond charge corrections, an alternative provided by LigParGen, is unlikely to lead to any improvement in entropy, given their mixed performance in calculating enthalpies of vaporization and density of liquids [52]. The more positive signed error for OPLS indicates that its entropies overall are larger than the GAFF entropies, implying that the combined intermolecular and intramolecular OPLS interactions are marginally weaker than GAFF. Contrary to this trend, the largest deviations between the force fields are OPLS being ∼20 J K${}^{-1}$ mol${}^{-1}$ lower than GAFF for ammonia and DMSO.

### 3.2. Entropy Components

To give deeper understanding into the values of the entropies, their six components in Equation (Equation 1) are illustrated in Figure 2 for the case of GAFF (Appendix A has numerical values for both force fields). Plotted in Figure 3 are the entropy components as a function of molecular mass, and Table 4 lists data for the lines of best fit. The first observation is the dominance of the molecular translational and rotational entropy, being more than half the total entropy for all but the largest molecules. $S_{M}^{\mathrm{transvib}}$ has a weak dependence on mass, deviating lower for systems at colder temperatures. $S_{M}^{\mathrm{rovib}}$ has a stronger mass-dependence and is lower for colder and linear molecules and those forming hydrogen bonds. One point to emphasise about our decomposition is that linear molecules in terms of united atoms, such as ethane or acetonitrile, have negligible rotational entropy about their long axis at the molecule level. The entropy about this axis including hydrogens is instead assigned to the united-atom level.

$S_{\mathrm{UA}}^{\mathrm{rovib}}$ is slightly smaller, making up about a quarter of the total. It primarily comprises the twisting of united atoms such as methyls (∼17 J K${}^{-1}$ mol${}^{-1}$) and hydroxyls (∼13 J K${}^{-1}$ mol${}^{-1}$) as well as hydrogen bending, such as in benzene, and thus relates more specifically to the number of hydrogens. As mentioned earlier, for linear molecules with two united atoms, it also includes the entropy of rotation about the long axis because this term would otherwise be zero without hydrogen. For example, for ethene $S_{M}^{\mathrm{rovib}}$ is smaller than for other molecules, and most of its $S_{\mathrm{UA}}^{\mathrm{rovib}}$ is rotational entropy about the long axis, leaving about 3 J K${}^{-1}$ mol${}^{-1}$ for internal motion. The remaining three terms are more variable and together make up about a quarter to a third of the total. The orientational term $S_{M}^{\mathrm{topo}}$ weakly increases with mass and is smaller for molecules with higher symmetry or those that form hydrogen bonds, which tend to reduce $N_{c}$. $S_{\mathrm{UA}}^{\mathrm{transvib}}$ mainly comprises dihedral vibration of united atoms and has a strong dependence on mass, as does the conformational term $S_{\mathrm{UA}}^{\mathrm{topo}}$, which is one of the smallest terms and only present for 13 liquids. The lines of best fit for each component indicate moderate predictability based on mass, but a thorough treatment is beyond the scope of this work. Comparing the force fields, GAFF has marginally higher molecular vibrational entropy (1.5 J K${}^{-1}$ mol${}^{-1}$) and higher $S_{\mathrm{UA}}^{\mathrm{topo}}$ (5.2 J K${}^{-1}$ mol${}^{-1}$) whereas OPLS has more $S_{\mathrm{UA}}^{\mathrm{rovib}}$ (2.2 J K${}^{-1}$ mol${}^{-1}$). Of the most extreme deviations, $S_{\mathrm{UA}}^{\mathrm{topo}}$ of GAFF is 14 J K${}^{-1}$ mol${}^{-1}$ higher than OPLS for 2-butoxyethanol and 12 J K${}^{-1}$ mol${}^{-1}$ higher for diethanolamine. Why this is so is revealed by an inspection of the probability distributions in Appendix A which indicate that the reduced $S_{\mathrm{UA}}^{\mathrm{topo}}$ for OPLS is because of stronger internal hydrogen-bonding. In more detail than looking at overall entropy, these trends imply that GAFF compared to OPLS has weaker intermolecular interactions, consistent with the charge over-polarisation of OPLS mentioned earlier [52], more evenly occupied conformations, and stronger intramolecular interactions, particularly relating to united-atom rotation.

A direct comparison of entropy components with 2PT for the 15 liquids in common [48] cannot be done because different OPLS force fields are used, but in general the 2PT molecular translational and rotational entropies are larger than the equivalent MCC terms, and the MCC terms become slightly larger upon inclusion of the orientational term. However, the three MCC united-atom terms are larger than the internal vibrational 2PT term, which in that work did not include a fluidicity term, as noted earlier [40]. However, later formulation of such a term [51] applied to ethane, methanol and hexane shows that the torsional fluidicity is only a few percent of the vibrational term, thus not being responsible for the difference with MCC.

### 3.3. Covariance Matrices and Coordination and Dihedral Distributions

Representative plots in Figure 4 show the force and torque covariance matrices respectively for the liquids using the GAFF force field (see Appendix A for all molecules). Similar to the combined force-torque matrices in earlier work [40], force covariance matrices show maximum auto-correlation along the diagonal and strong anti-correlation for bonded atoms. Correlations between more distant atoms are only evident for more rigid molecules, consistent with their lower vibrational entropy. Torque covariance matrices have weak correlations, most ranging from negligible up to a tenth of the diagonal self-correlation, consistent with the mean-field approximation made for united-atom rotation. Only very rigid molecules such as ethene display large correlations but their associated entropy is very small. Molecule-level matrices are not shown, being near-purely diagonal.

Representative $p(N_{c})$ distributions of five liquids with the GAFF force field are shown in Figure 4 (see Appendix A for all liquids). As expected for liquids, these distributions are broad and roughly Gaussian, most peaking between $N_{c}=5$ and 10. As Equation (Equation 4) makes clear, larger coordination brings about larger orientational entropy. The outliers with higher coordination are the six-membered rings such as cyclohexane, piperidine and 1,4-dioxane, and carbon dioxide, versus the hydrogen-bonded molecules whose hydrogen-bonds bring about more directed interactions and lower $N_{c}$, such as methanol, diethanolamine and octanol, the last of which is slightly liquid-crystalline.

The dihedral probability distributions $p_{i}$ are given in Appendix A for the 13 molecules with united-atom dihedrals. Of the 11 molecules with more than one dihedral, the correlation matrix brings about only a small reduction in entropy relative to the ideal value for independent dihedrals, indicating that conformations in these non-ring systems are weakly correlated. The largest reductions are −4.2 and −1.0 J K${}^{-1}$ mol${}^{-1}$ for OPLS and GAFF triethylamine, followed by −1.0 and −0.4 J K${}^{-1}$ mol${}^{-1}$ for OPLS and GAFF 2-butoxyethanol and −0.6 J K${}^{-1}$ mol${}^{-1}$ for both OPLS and GAFF octanol. However, for the ring molecules, such as cyclohexane, piperidine and 1,4-dioxane, which have six fully correlated dihedrals the method correctly picks out their two possible conformers as eigenvectors with eigenvalues according to their probability, with all other eigenvalues being zero. In the short timescale here, only a few molecules in each system convert to the other conformer. Achieving equilibrium is unnecessary for cyclohexane and 1,4-dioxane because both conformers are identical and contribute no entropy. However, the equatorial and axial conformers of piperidine are distinct, with the equatorial hydrogen on the nitrogen being lower in energy by 1.7 K J mol${}^{-1}$ [72], which would increase entropy by ∼5 J K${}^{-1}$ mol${}^{-1}$.

## 4. Discussion

We have extended our MCC method to calculate entropy for a much broader range of 56 liquids than the 14 liquids studied previously [40]. To emphasise the advantages of MCC, it is simple in its theoretical formulation, informative by giving an entropy decomposition over all degrees of freedom, rapidly convergent in the number of simulation frames required, scalable to large systems with its multiscale formulation, near-general and applicable to a huge range of molecular systems, and accurate to the level of the thermal energy $k_{B}T$ for the liquids studied here.

Of the improvements incorporated in this work, the first is the recognition that the force-halving arising from the mean-field cell approximation should not be applied to bonded united atoms because of their strong correlation. This leads to lower entropies than previously [40], which is especially important for the larger molecules such as toluene or cyclohexane. The good agreement obtained earlier for single flexible molecules [39] was likely obtained due to a cancellation of errors, with the missing rotational entropy of united atoms offsetting the larger entropy due to force halving in the force covariance matrix. Nonetheless, averaged out correlations in the force and torque covariance matrices owing to conformational fluctuations may account for MCC entropies being lower than experiment for larger molecules. A more minor modification from previous work [40] relates to the use of separate force and torque covariance matrices, rather than a combined force-torque covariance matrix, owing to weak correlation between forces and torques, a change which improves the computational efficiency of the method. This work shows that subunit torques are weakly correlated in most cases, meaning that even the torque covariance may be unnecessary.

The second principle improvement is the correlation matrix to account for the correlation of dihedral conformations by expressing the conformational distribution in terms of a basis of conformers. A key feature of the correlation-matrix method is that it efficiently scales to large systems, with matrix size increasing as $N_{\mathrm{dih}}^{2}$. Considering each conformer separately goes exponentially as $3^{N_{\mathrm{dih}}}$ and would become unfeasible beyond $N_{\mathrm{dih}}>10$. The traditional approach using correlations in continuously valued dihedral angles has an even worse exponential dependence and goes as $N_{\mathrm{bin}}^{N_{\mathrm{dih}}}$, where $N_{\mathrm{bin}}$ is the number of bins. This is already problematic for $N_{\mathrm{dih}}>2$, but it can be somewhat relieved by nearest-neighbour methods [16,22,23,24]. It is reasonable to assume that dihedral correlations need only be considered for local energy wells rather than for the numerical value of the dihedral, given that this correlation is unlikely to change on the timescale of molecular vibration.

A third issue to consider in future work is the multiscale approximation in how different levels of hierarchy are defined, how to avoid the double-counting of entropy between different levels of hierarchy, and how to streamline the theory further so that it is essentially equivalent at every level of hierarchy to maximise generality. Ideally, the determination of each level would be automated and dynamic, adjusting to the level of order in the system. Care is needed to ensure that the translational or rotational entropy duplicates that at the higher level for every level of hierarchy so that it is cleanly removed. The theory for vibrational entropy is already quite general for any level of hierarchy, while the topographical terms require more work to fuse Equations (Equation 4) and (Equation 8) into the same formulation. This would involve generalising the orientational entropy to be non-ideal so that orientations have different weightings according to the orientations of the neighbouring molecules, as has been already studied for water with its strongly directional hydrogen bonds [35,37,42,44]. Including the united-atom orientational entropy could be extended to other molecules such as alcohols and amines. Nonetheless, the framework is in place to scale the method to simulated systems of greater complexity.

## 5. Conclusions

We have presented the multiscale cell correlation method to calculate the entropy of 56 molecular liquids from molecular dynamics simulations. The entropies are in excellent agreement with experiment for the OPLS and GAFF force field, with GAFF performing slightly better. Agreement is better than that of the 2PT method, which can also calculate the entropy of molecular liquids. The components of entropy give an insightful and intuitive understanding of the values obtained. With suitably chosen levels of hierarchy, the method is readily scalable to larger and more complex systems.

## Figures and Tables

**Figure 1 entropy-21-00750-f001:**
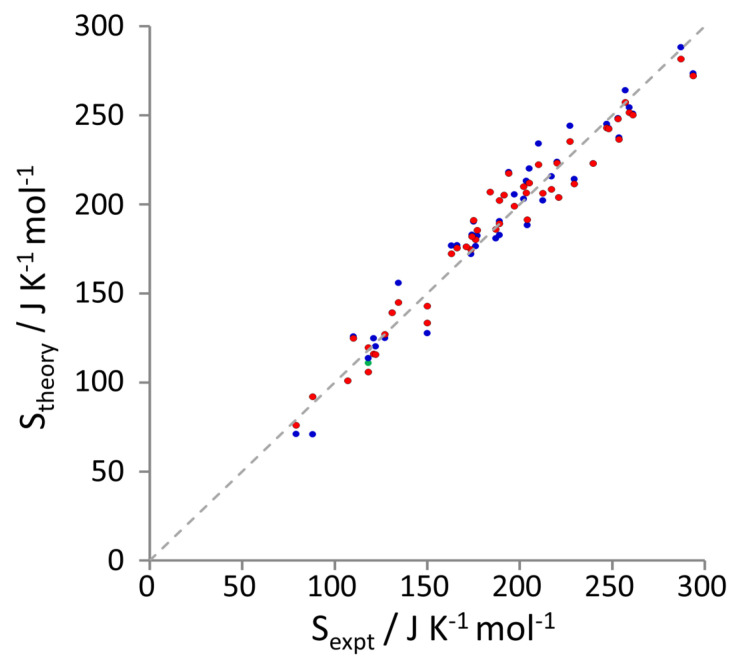
Multiscale cell correlation (MCC) entropy values versus experiment for OPLS (blue), GAFF (**red**), and TraPPE (**green**), together with the line of perfect agreement (**dotted**).

**Figure 2 entropy-21-00750-f002:**
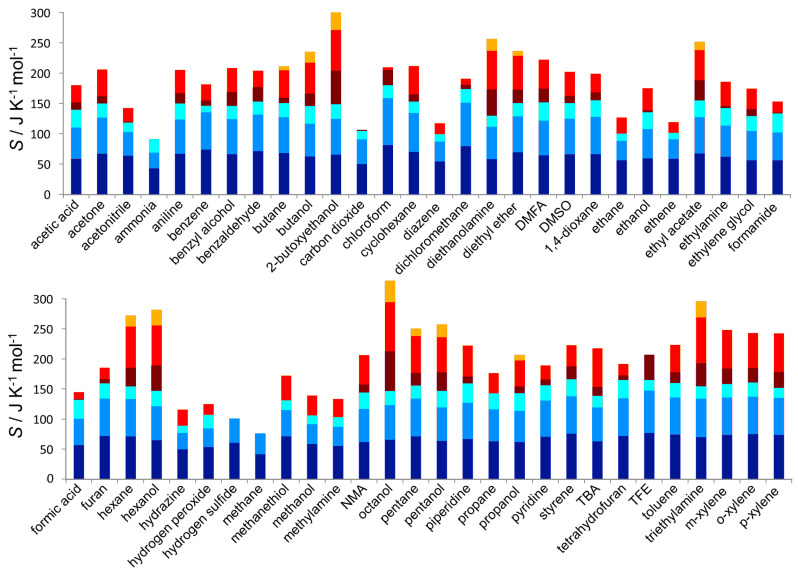
MCC entropy components for GAFF (bottom to top): molecular-translational (dark blue), molecular rotational (blue), molecular topographical (cyan), united-atom translational (dark red) united-atom rotational (red), and united-atom topographical (orange).

**Figure 3 entropy-21-00750-f003:**
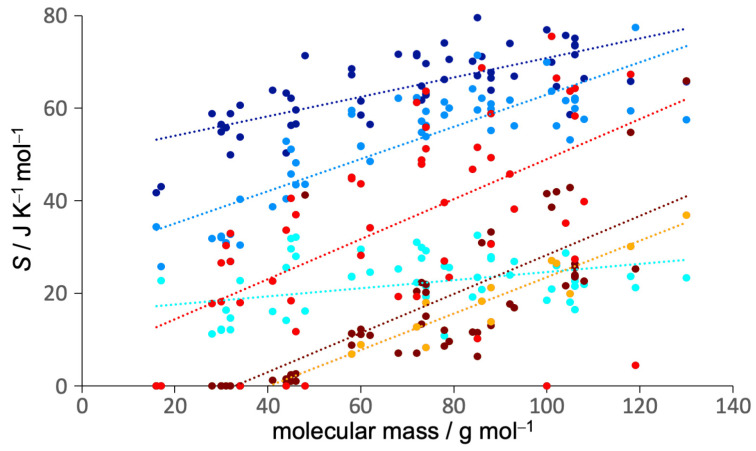
MCC entropy components for GAFF versus molecular mass for all liquids. The colouring is as in Figure 2.

**Figure 4 entropy-21-00750-f004:**
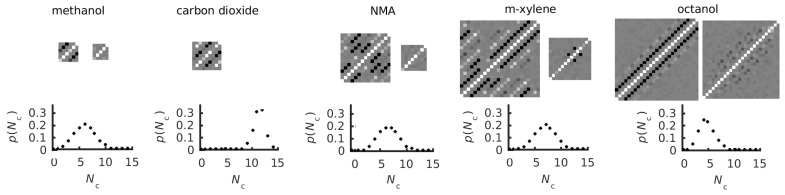
Panels for five representative liquids (GAFF) illustrating the UA force (left) and torque (right) covariance matrices and coordination-number probability distributions $p(N_{c})$ (lower). For the matrices, white and black represent correlations of 1 and −1, respectively, with grey in between. The matrix origin is at the lower left.

**Table 1 entropy-21-00750-t001:** Boiling Temperature of Liquids [63] that are Gaseous at Ambient Conditions.

Liquid	*T*/K	Liquid	*T*/K	Liquid	*T*/K	Liquid	*T*/K
ammonia	240	ethane	185	hydrogen sulfide	213	methylamine	267
butane	272	ethene	170	methane	112	propane	231
carbon dioxide	220 ${}^{a}$	ethylamine	291	methanethiol	279	TFE	197
diazene	275						

^*a*^ Pressure is 5.99 bar.

**Table 2 entropy-21-00750-t002:** Statistical Data for MCC and 2-Phase Thermodynamics (2PT) versus Experiment.

Data Set (Number of Liquids)	$\langle |S-S_{\mathrm{expt}}|\rangle$/J K${}^{-1}$ mol${}^{-1}$	$\langle S-S_{\mathrm{expt}}\rangle$/J K${}^{-1}$ mol${}^{-1}$	Slope	Y-Intercept/J K${}^{-1}$ mol${}^{-1}$	$R^{2}$	Zero-Intercept Slope
MCC OPLS ${}^{a}$ (46)	9.8	0.6	0.94	11.7	0.95	1.00
MCC GAFF (50)	8.7	−0.3	0.93	13.0	0.96	0.99
2PT OPLS ${}^{b}$ (12)	15.5	−15.6	1.05	−25.3	0.84	0.92
2PT GAFF (14)	28.0	−24.4	0.97	−19.5	0.55	0.87
MCC OPLS ${}^{a}$ (12)	4.9	2.3	0.87	26.7	0.89	1.01
MCC GAFF (14)	7.6	4.0	0.93	16.5	0.93	1.02

${}^{a}$ OPLS with 1.14*CM1A charges [52]; ${}^{b}$ OPLS with charges optimised to liquid-phase properties [62].

**Table 3 entropy-21-00750-t003:** Entropy by Experiment, MCC and 2PT (J K${}^{-1}$ mol${}^{-1}$).

Liquid	Experiment ${}^{a}$	MCC		2PT [48]	
		OPLS	GAFF	OPLS	GAFF
acetic acid	158, 194	177	180	147	128
acetone	200	202	206	198	187
acetonitrile	150		143		145
ammonia	87 ${}^{b}$	71	92		
aniline	191, 192	205	205		
benzene	173, 175	183	182	172	161
benzyl alcohol	217	216	208		
benzaldehyde	221	204	204		
butane	227, 230, 231	214	212		
butanol	226, 228	244	235		
2-butoxyethanol		293	301		
carbon dioxide	118 ${}^{c}$	111 ${}^{d}$	106	112 ${}^{d}$	
chloroform	202 ^e^	203	210	193	226
cyclohexane	204, 206	220	212		
diazene	121	125	116		
dichloromethane	175	190	191		
diethanolamine		248	256		
diethyl ether	253, 254	237	236		
DMFA		214	222		
DMSO	189	183	202	164	159
1,4-dioxane	197	206	199	179	159
ethane	127	125	127		
ethanol	160, 161, 177	177	175	141	127
ethene	118	114	120		
ethyl acetate	259	254	252		
ethylamine	189 ${}^{f}$	181	185		
ethylene glycol	167, 180	172	175	141	121
formamide		151	153		
formic acid	128, 132, 143	156	145		
furan	177	181	186	167	157
hexane	290, 295, 296	273	272	251 ${}^{g}$	
hexanol	287	288	281		
hydrazine	122	120	116		
hydrogen peroxide	110 ${}^{h}$	126	125		
hydrogen sulfide	106 ${}^{i}$		101		
methane	79 ${}^{j}$	73	78		
methanethiol	163	177	172		
methanol	127, 130, 136	139	139	117 ${}^{g}$, 122	109
methylamine	150	128	133		
NMA		205	206	181	168
octanol		335	331		
pentane	259, 263	251	250		
pentanol	255, 259	264	257		
piperidine	210	234	222		
propane	171	176	176		
propanol	193, 214	213	206		
pyridine	178, 179, 210	191	189		
styrene	238, 241	223	223		
TBA	190, 198	218	217		
tetrahydrofuran	204	188	192	196	159
TFE	184		207	195	185
toluene	219, 221	224	223	204	190
triethylamine	309 ${}^{f}$	292	295		
m-xylene	252, 254	248	248		
o-xylene	246, 248	245	245		
p-xylene	244, 247, 253	243	243		

${}^{a}$ Reference [63] Experimental errors <1 J K${}^{-1}$ mol${}^{-1}$ ; ${}^{b}$ Reference [64]; ${}^{c}$ References [50,69]; ${}^{d}$ TraPPE force field [58]; ^*e*^ Reference [65]; ${}^{f}$ Derived in Appendix A using References [63,70]; ${}^{g}$ Reference [51]; ${}^{h}$ Reference [67]; ${}^{i}$ Reference [68]; ${}^{j}$ Reference [66].

**Table 4 entropy-21-00750-t004:** Lines of Best Fit for the Entropy Components versus Molecular Mass.

Component	Slope/J K${}^{-1}$ g${}^{-1}$	Y-Intercept/J K${}^{-1}$ mol${}^{-1}$	$R^{2}$	Component	Slope/J K${}^{-1}$ g${}^{-1}$	Y-Intercept/J K${}^{-1}$ mol${}^{-1}$	$R^{2}$
$S_{M}^{\mathrm{transvib}}$	0.21	50	0.54	$S_{\mathrm{UA}}^{\mathrm{transvib}}$	0.42	14	0.63
$S_{M}^{\mathrm{rovib}}$	0.35	28	0.70	$S_{\mathrm{UA}}^{\mathrm{rovib}}$	0.43	6	0.34
$S_{M}^{\mathrm{topo}}$	0.09	16	0.13	$S_{\mathrm{UA}}^{\mathrm{topo}}$	0.39	16	0.87

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
