# Peer review of "Entropy of Simulated Liquids Using Multiscale Cell Correlation"

_entropy, 2019, doi:10.3390/e21080750_

Round 1

Reviewer 1 Report

The manuscript by Ali and coworkers presents a large-scale comparison of entropies of different liquids, calculated by the newly developed multiscale cell correlation (MCC) method against their experimental counterparts. In addition, the authors compare their values with those of another computational method, 2PT. The MCC method is an updated variant of the approach developed by the lead author Richard Henchman and coworkers and is one of the most rigorous and conceptually cleanest approaches around for the treatment of the entropy of liquids. The article is clearly written and is an important milestone for further developments, and for these reasons I recommend its publication in Entropy, after the following minor comments have been taken into the account.

1. The authors should elaborate more on the good correlation between MCC and experimental entropies. In this context, it is necessary to discuss how susceptible MCC values are to certain simulation parameters. In particular, how do, e.g., the short non-bonded cut-off of 8A, the particle-mesh Ewald summation (and its possible perturbation of structural states) and potentially not realistic simulation temperatures (i.e. simulation temperatures in the force field not corresponding to the real value) influence MCC values. It would be very instructive for a general reader to understand how come, despite all these potential problems, the simulations yield such ultimately good match with experiment.

2. The authors should comment some more on the variability of experimentally obtained entropies, which for some chemicals is more than 20%, and how this relates to the comparison of MCC vs. 2PT and the comparatively similar difference seen there.

3. Furthermore, concerning the difference of calculated entropies using the OPLS and the GAFF force fields, the authors should comment on the higher variability between the two in the 2PT method as compared to the MCC method.

4. Figures 3,4 and 5 should be moved to Supplementary Information as they are not too informative as is. At the same time, Figure 2 has the potential to be more informative if presented as a non-stacked histogram and/or as individual graphs showing the dependence of different components on size.

5. Line 87, it should be “closer”.

6. Line 396, it should “experiment”.

7. Line 398, it should be “give”.

Author Response

See attached pdf.

Reviewer 2 Report

Journal Name: Entropy

Manuscript ID Number: 550050

Manuscript Title: Entropy of Simulated Liquids Using Multiscale Cell Correlation

Review Comments:

In this work authors present multiscale cell correlation method to calculate the entropy of liquids from molecular dynamics simulations. There are few issues to be handled for further review prior to the final decision that can be made on the acceptance of this manuscript and these are provided below.

1.     How id the authors deal with the force field parametization? Please clarify this in the revised version of the manuscript.

2.     Figure 1 might have a second image panel and show the percentage deviations from ideal and expt entropy values for both the OPLS, GAFF and TraPPE methods.

3.     What is the uncertainty in the NIST values for the extracted values from the Chemistry Webbook? Please include this under the table 2 as per the NIST style or the journal standards.

4.     I think figures 3-4-5 can be moved to the ESI and authors can leave some representative images in this main text in order to make it more clear.

Author Response

See attached pdf.

Reviewer 3 Report

The manuscript “Entropy of Simulated Liquids Using Multiscale Cell Correlation” by Ali et al. describes an extension of multiscale cell theory to allow for entropy estimates of more complex molecular liquids from equilibrium molecular dynamics simulations. Specific improvements are related to the treatment of uncorrelated and correlated degrees of freedom of the molecules with explicit correlation for dihedral angles. 

The overall performance as quantified by the agreement between simulated and experimental absolute entropies of liquids is very promising and has been obtained with two distinct force field parameterizations. The described procedure can be applied to a wide array of liquids as shown by the extensive test set, with some remaining problems for highly structured liquids with pronounced hydrogen bond networks, e.g. water, for which specialized terms have been introduced in earlier applications. This is properly discussed in the closing discussions. 

I have a couple of comments regarding the clarity of the manuscript, the benefit of absolute entropies over entropy differences, and the comparison to experimental data.

1)  Clarity

A few passages in the manuscript are difficult to decipher, e.g.:

“The first improvement here in MCC is a more appropriate application of the mean-field cell approximation to the weakly correlated degrees of freedom and not to the correlated degrees of freedom as had been done in previous work [39,40]. Such strong correlations for the bonded atoms invalidate the cell approximation and in any case can be accounted for in the force covariance matrices.”

It is not clear what the correlated degrees of freedom are, which are mentioned in the first sentence and are now treated differently, without reading the cited references. However, the second sentence directly refers to them.

Equation 8 seems to follow the common formulation of a Shannon entropy S = -\int p(x) ln[p(x)] dx, while equations 4 and 5 in their current form do not. A short explanation would be very helpful, especially because the notation/formulation used in Reference 40, which is cited in the context of these equations, is slightly different.

Line 105: What is a united atom with no hydrogen? It seems that the united atom concept does not apply here.

In line 272, the authors conclude based on their results that the interactions in OPLS simulations are weaker than in simulations based on GAFF. However, on line 304, they state that the intermolecular interactions (which I assume would contribute to most of the entropy) are stronger for OPLS. I find this quite confusing.

Line 178: The authors seems to indicate that dihedral angles, which have “only one conformation by symmetry, such as a methyl group” can be ignored. However, each conformation, even if symmetrically equivalent, describes a valid microstate, which contributes to the entropy. Maybe the authors can clarify this point as it seems the entropy of methyl group rotations has been considered here, at least if this is meant by the “twisting of groups such as methyls” in line 287.

2) Absolute entropies

In their comparison to the 2PT approach, the authors indicate the increased accuracy of the MCC method. However, to describe thermodynamic processes, it seems that describing entropy differences accurately would be more important. The latter would be described by the unconstrained slope in Table 3, which, in my interpretation of the provided statistical measures, would allow for a constant offset. Here, 2PT seems to be at least competitive to the MCC. This mentioned briefly, but should be discussed in more detail. Especially, since the agreement of the absolute MCC entropy with experiments is clearly better for the studied liquids, it would be very helpful to provide examples of applications for which accurate estimates of absolute entropies are required. 

3) Comparison to experiments

Comparing observables from molecular dynamics simulations with approximate formalisms and comparing the results with experiments has intrinsic problems, as one automatically calculates the combined error of the force field model, the neglect of quantum effects and the new methodology. Excellent agreement with experiments may therefore result from error compensation between the classical force field simulation model and the analysis procedure. This should be discussed in more detail by the authors. 

The more meaningful comparison presented here is the one between the 2PT analysis and MCC for simulations with GAFF, as both approximate methods, are applied to simulations of the same models. As discussed above, unless an application requires explicit knowledge of the absolute entropy, constant offsets between the predictions should be allowed, which would cancel whenever entropy differences are calculated (usually between thermodynamic states, not chemical species, but the general concept still applies). 

It would be ideal, if a feasible gold standard approach existed to obtain the molecular entropy of liquids from simulations, but as pointed out by the authors, this is not currently the case.

Author Response

See attached pdf.

Round 2

Reviewer 1 Report

The authors have adequately addressed all of my concerns.

Reviewer 3 Report

The authors have addressed my previous comments and the manuscript is publishable.